# Solar Ultraviolet Radiation in Pretoria and Its Relations to Aerosols and Tropospheric Ozone during the Biomass Burning Season

D. Jean du Preez [1,2,*], Hassan Bencherif [1,3], Thierry Portafaix [1], Kévin Lamy [1] and Caradee Yael Wright [2,4]

1 LACy, (UMR 8105, CNRS, Université de La Réunion, Méteo-France), 97744 Saint-Denis de La Réunion, France; hassan.bencherif@univ-reunion.fr (H.B.); thierry.portafaix@univ-reunion.fr (T.P.); kevin.lamy@univ-reunion.fr (K.L.)
2 Department of Geography, Geoinformatics and Meteorology, University of Pretoria, Pretoria 0002, South Africa; caradee.wright@mrc.ac.za
3 School of Chemistry and Physics, University of KwaZulu-Natal, Durban 4041, South Africa
4 Environmental and Health Research Unit, South African Medical Research Council, Pretoria 0001, South Africa
* Correspondence: dupreez.dj@tuks.co.za

**Abstract:** Biomass burning has an impact on atmospheric composition as well as human health and wellbeing. In South Africa, the biomass burning season extends from July to October and affects the aerosol loading and tropospheric ozone concentrations which in turn impact solar ultraviolet radiation (UVR) levels at the surface. Using ground-based observations of aerosols, tropospheric ozone and solar UVR (as well as modelled solar UVR) we investigated the impact of aerosols and tropospheric ozone on solar UVR in August, September, and October over Pretoria. Aerosol optical depth (AOD) and tropospheric ozone reached a peak between September and October each year. On clear-sky days, the average relative difference between the modelled and observed solar Ultraviolet Index (UVI) levels (a standard indicator of surface UVR) at solar noon was 7%. Using modelled UVR—which included and excluded the effects of aerosols and tropospheric ozone from biomass burning—aerosols had a larger radiative effect compared to tropospheric ozone on UVI levels during the biomass burning season. Excluding only aerosols resulted in a 10% difference between the modelled and observed UVI, while excluding only tropospheric ozone resulted in a difference of −2%. Further understanding of the radiative effect of aerosols and trace gases, particularly in regions that are affected by emissions from biomass burning, is considered important for future research.

**Keywords:** aerosol optical depth; Africa; air pollution; atmospheric science; environmental health; ozone; UV

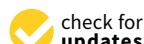

## 1. Introduction

As solar ultraviolet radiation (UVR) passes through the atmosphere it interacts with gases and particles which absorb, reflect, or scatter the incoming solar UVR. Solar UVR is classified into three bands: UVA (315–400 nm), UVB (280–315 nm), and UVC (100–280 nm) where the absorption of solar UVR by stratospheric ozone increases within the UVB spectrum [1]. As a result, surface solar UVR is decreased significantly at shorter wavelengths [1]. In the troposphere, solar UVR is further attenuated by tropospheric ozone, sulfur dioxide, aerosols, and clouds [2]. Other factors such as altitude, solar zenith angle, and albedo effect solar UVR levels at the surface [1,3].

Atmospheric aerosols have direct and indirect effects on the Earth's radiation budget. The direct effect of aerosols on the radiation budget is due to the scattering and absorption of UVR by aerosols [4], while the indirect effect of aerosols is due to the formation of clouds as aerosols can act as cloud condensation nuclei [5]. The radiative effect of aerosols is determined by their size, distribution, and optical properties [6]. Atmospheric aerosols can

be from natural or anthropogenic sources. Natural sources include dust storms, volcanic eruptions, sea salt spray, and biomass burning. Biomass burning emissions can also be anthropogenic along with other human-made emissions such as those from vehicles and industries. [7].

Biomass burning is one of the largest contributors to tropospheric aerosol loading where these aerosols also have a significant radiative effect [8,9]. In Southern Africa, the main biomass burning region is located over the north and eastern parts of South Africa and Mozambique. The biomass burning season reaches an annual peak at the end of the dry season between August and September [10]. Emissions from biomass burning are transported westward to the Atlantic Ocean as well as eastward towards the Indian Ocean [11]. During spring, the semi-permanent Indian Ocean anticyclone and easterly winds transport aerosols over southern Africa [10,12,13]. As a result, the radiative effect of aerosols is not limited to the burnt area and the resulting plumes can have an impact thousands of kilometers away [14,15].

The emissions released during the biomass burning process (and other pollution-generating activities) include ozone precursors such as nitrogen oxides (NOx), carbon monoxide (CO), and volatile organic compounds (VOCs) [16]. These gases react with sunlight to form ozone in the troposphere [17]. The formation of tropospheric ozone is dependent on the concentration of ozone precursors as well as temperature and humidity [18]. Tropospheric ozone has a short lifetime and is a greenhouse gas. It also impacts human health, vegetation, and crop yields [19]. The emission of ozone precursors is not the only source of tropospheric ozone. Stratosphere-troposphere exchange (STE) can result in an increase in tropospheric ozone levels due to the higher levels of ozone present in the stratosphere [17].

Due to the synoptic-scale circulation pattern, aerosol loading and tropospheric ozone levels over Pretoria are affected by biomass burning emissions. This study aimed to investigate the effect of aerosols and tropospheric ozone on surface UVR levels over Pretoria during the biomass burning season. Furthermore, a comparison of observed and modelled UVR was made to determine the separate and combined influence of aerosols and tropospheric ozone from biomass burning emissions on solar UVR. These findings are useful to improve the understanding of the radiative effect of tropospheric ozone and aerosols on solar UVR levels at the surface. In the data and methods section, the study area, data, instrumentation, and methods are described. The results from the data analysis show the seasonal cycle of tropospheric aerosols, ozone, and UVR as well as a case study and model simulations to demonstrate the radiative effect of tropospheric aerosols and ozone.

## 2. Data and Methods

Pretoria is situated on the inland plateau of South Africa in the Gauteng province at approximately 1300 m above sea level (Figure 1). The city experiences cool, dry winters and long, hot summers with rainfall occurring during the summer months. Pretoria was selected as the study area due to the known high levels of tropospheric ozone related to industrial activities [12,20] and the impact of the biomass burning season on aerosol distribution [21]. Data were collected from three stations, namely the South African Weather Service (SAWS) Bolepi House, SAWS Irene, and the Council for Scientific and Industrial Research (CSIR) head office due to their relative proximity to one another. The three different stations were located between 1322 m and 1529 m above sea level. The CSIR and Irene stations are approximately 18 km apart, while Bolepi House is approximately 6 km from the CSIR.

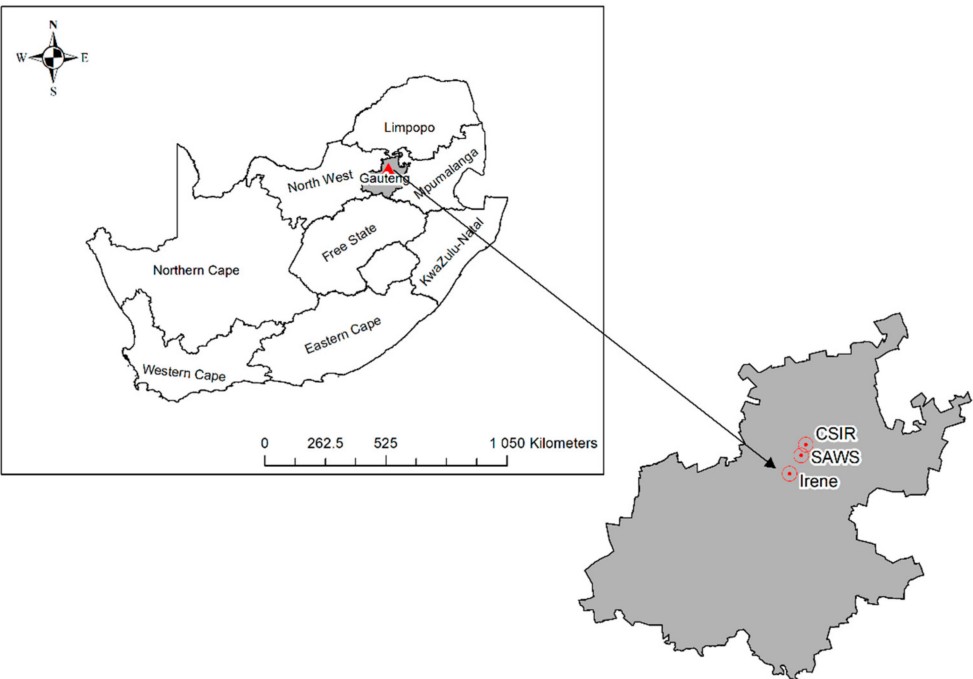

**Figure 1.** Map showing the location of Pretoria in South Africa and the three stations where data were collected from the Council for Scientific and Industrial Research (CSIR) head office, South African Weather Service (SAWS) Bolepi House and SAWS Irene weather stations, respectively.

### 2.1. Aerosol Data from the CSIR Station

A Cimel sun photometer is located at the CSIR head office in Pretoria (25.76° S, 28.28° E) at 1 449 m above sea level (Figure 1) and is part of the Aerosol Robotic Network (AERONET). The sun photometer has a spectral range of 340–1640 nm with eight spectral bands (340, 380, 440, 500, 675, 875, 1020, and 1640 nm) [22]. Details on the instrument, calibration and error estimation are published in Dubovik et al. [23]. Aerosol Optical Depth (AOD) is an indication of the distribution of aerosols in a column of air. Additional observations at 935 nm are used to estimate columnar water vapour [24]. The Ångström Exponent (AE) can be calculated using Equation (1) and can be used to estimate the size distribution of aerosols from spectral AOD observations [25]. AE values less than 1.0 indicated that coarse particles such as desert dust were dominant while AE values greater than 1.0 indicated the fine particles, such as smoke and sulfates, were dominant [25–28]. Equation (1) follows:

$$\alpha = -\frac{\log \frac{\tau_{\lambda 1}}{\tau_{\lambda 2}}}{\log \frac{\lambda_1}{\lambda_2}} \tag{1}$$

where $\alpha$ is the AE, $\tau_\lambda$ is the AOD at the first and second wavelengths ($\lambda$), respectively.

The AERONET inversion algorithm is used to provide aerosol optical properties such as single scattering albedo (SSA) which is derived from direct and diffuse radiation measurements from sun photometers [29]. The inversion algorithm assumes that the vertical distribution of the aerosols is similar to global models, particles are partitioned into spherical and non-spherical and it accounts for the gaseous absorption by ozone, nitrogen dioxide, and water vapor. Furthermore, the algorithm provides information on the quality of the output parameters produced [30]. SSA is an important factor related to the radiative effect of aerosols and represents the ratio between the scattering and extinction efficiencies of aerosols [31]. The level two daily average data of AOD (340 nm) and AE (340–440 nm) were obtained from the AERONET website (aeronet.gsfc.nasa.gov/) for the period from 1

July 2011 to 31 May 2018 (inclusive) and were used to calculate the monthly averages and standard deviations of AOD and AE.

### 2.2. Tropospheric Ozone Data from the Irene Station

Since 1998, ozone soundings have been conducted at the SAWS Irene weather station (25.9° S, 28.2° E) in Pretoria (Figure 1). The station is 1 529 m above sea level and operates within the Southern Hemisphere Additional Ozonesondes (SHADOZ) network [32–34]. The ozone soundings were conducted twice a month using an electrochemical cell comprising a cathode and anode cell. The solution of buffered potassium iodide (KI) and saturated solution of KI was used in the cathode and anode cell, respectively. An ion bridge connected the cells which allowed for electrons to flow between the two chambers and ozone was measured using the iodine/iodide electrode reactions. An interface board was connected to a radiosonde to transmit data regarding cell current, pump temperature, ambient temperature, pressure, and relative humidity [33].

The data for 1998–2018 were obtained from the SHADOZ website (tropo.gsfc.nasa.gov/shadoz/index.html). Between 1998 and 2018, 369 ozonesondes were launched, however, no ozonesondes were launched from 2008 to 2012. Within this dataset, there are gaps when launching ozonesondes was not possible due to various reasons. The raw data were obtained at two-second intervals and the ozone concentrations were averaged over 100 m intervals from the surface to the burst altitude (± 30 km) of the balloon. The tropospheric ozone column was calculated, in Dobson Units (DU), by integrating the vertical ozone profile from the surface to the lapse-rate tropopause (LRT). The LRT is defined as the lowest level at which the temperature lapse is less than 2 K km$^{-1}$ for at least 2 km [35]. Each ozone sounding was used to determine the average tropospheric column and average ozone profile for the respective month.

### 2.3. Observed UVB Data for Pretoria

The SAWS had a surface UVB radiation monitoring station at their headquarters, Bolepi House, in Pretoria (25.81° S, 28.26° E) [36]. A Solar Light 501 UVB radiometer with a spectral wavelength of 280–315 nm was used to measure radiation at hourly intervals. The instrument provided an analogue voltage output proportional to the measured radiation [37] that was given in Minimal Erythemal Dose (MED) units. One MED was approximately 210 Jm$^{-2}$, where MED is a metric used to express the minimal erythemal dose required to induce erythema (also known as sunburn) [38]. The MED units were converted to UV Index (UVI), a standard indicator of UVR levels [1] using Equation (2) [39]:

$$\text{UVI} = \frac{210\left[J.m^{-2}\right] \times 40\left[m^2.W^{-1}\right]}{3600\left[s/h\right]} \tag{2}$$

To correct the instrument-weighted UVB radiation to the erythemal weighted UVR spectrum (280–400 nm) a correction factor was applied [40]. The correction factor used satellite observed total column ozone to correct for the spectral and angular response of the instrument. A comparison between observed UVB and satellite-derived UVB showed that there was a moderately strong correlation between these variables at the Pretoria station [41]. The instrument was last calibrated in 2013 by the Deutscher Wetterdienst (DWD). The long period since calibration is noted as a limitation and may be a source of uncertainty in the UVB radiation data. Hourly data from 1 January 2009 to 30 April 2018 (inclusive) were obtained from this station for this study.

Clouds result in large spatial and temporal variability of surface UVR radiation levels. To remove the effect of clouds on observed UVR, clear-sky days were determined using a clear-sky determination method [42,43]. To do so, three steps were used. The first step calculated the correlation between UVR values before solar noon and after solar noon. The values after solar noon were reversed so that the UVR values with similar SZAs were correlated. When the correlation was below 0.8, the day was defined as cloudy. The second step tested for monotonic increases and decreases in UVR values before and after solar noon,

respectively. If the increases and decreases in UVR values were not monotonic, the days were defined as cloudy. Lastly, the data were used to determine the monthly hourly average values for all-sky conditions. If the UVR at solar noon was 1.5 standard deviations below the average, then the day was defined as cloudy. The clear-sky determination methods identified 1190 clear-sky days during the observation period (2009 to 2018).

From the hourly data, the monthly hourly averages, and standard deviations of UVR at solar noon were calculated only using days with a complete observation record. The observed clear-sky solar noon UVR values were compared to the modelled clear-sky UVR values to show that there was a consensus between the observed and modelled data (Section 2.4).

### 2.4. Modelled UVR over Pretoria

To investigate the radiative effect of aerosols and tropospheric ozone over Pretoria, the Tropospheric Ultraviolet-Visible (TUV) radiative transfer model version 5.3 [44] was used to model clear-sky UVI at Bolepi House. Using pseudo-spherical, eight-stream discrete ordinates to solve the radiative transfer algorithm [45] the model accounted for the scattering and absorption of UVR by gases and particles as it passed through the atmosphere [46]. The model calculated parameters including radiance, weight spectral integrals for specific wavelength bands and biologically-active irradiance, i.e., UVI. Input data for the model included total column ozone, total column nitrogen dioxide, climatological ozone, and temperature profiles, AOD (340 nm), AE (340–440 nm), SSA, and altitude.

In this study, daily total column ozone and nitrogen dioxide observations from the Ozone Monitoring Instrument (OMI) were used [47,48]. McPeters and Labouw [49] derived monthly averaged zonal ozone and temperature profile for different latitude regions between 0 and 60 km using data from ozonesondes for the troposphere, the Stratospheric Aerosol and Gas Experiment (SAGE II) [50] and Microwave Limb Sounder (MLS) instrument for the stratosphere [49]. The ozone and temperature climatological profiles for $25°$ S included data from the ozonesondes launched at Irene and the MLS instrument provided measurements down to the thermal tropopause. The input data for the aerosol optical properties included daily averages of AOD and AE which were obtained from the AERONET sun photometer at the CSIR and the aerosol distribution described by Elterman, 1968 [51]. Monthly SSA (550 nm) data were obtained from the Max-Planck Version 2 (MACv2) aerosol climatology [52] as the inversion algorithm did not provide sufficient data points. The MACv2 aerosol climatology used ground-based measurements and modelling to create global data fields.

Data on the aerosol optical properties (AOD and AE) were only available for 1 July 2011 to 31 May 2018. Therefore, the TUV model was used to calculate the clear-sky UVI levels for this period during which there were 1215 days which had complete input data required for the TUV simulations. The simulated UVI was compared to the observed clear-sky UVI by calculating the relative difference for each month using Equation (3):

$$\text{Relative difference} = 100 \times (\text{UVI}_{\text{TUV}} - \text{UVI}_{\text{obs}}) \div \text{UVI}_{\text{obs}} \tag{3}$$

where TUV represents the modelled UVI and Obs the observed UVI.

### 2.5. Effect of Aerosol and Ozone on UVR over Pretoria

Biomass burning affects atmospheric composition through the release of aerosols and ozone precursors. Approximately 31% of AOD observations were one standard deviation above the respective monthly averages from August to October (2011 to 2017). To investigate the radiative effect of aerosols, the observed clear-sky and model UVI values were compared to AOD observations during this period.

To assess the sensitivity of surface solar UVI to aerosols and tropospheric ozone during the biomass burning season (August to October) four TUV scenarios were simulated and compared to the ground-based UVI observations. The background level of aerosol optical properties (AOD, AE, and SSA) and tropospheric ozone were determined by averaging

the monthly values from November to July when AOD and tropospheric ozone were at their lowest. The aerosol optical properties and tropospheric ozone between August and October were determined to be the biomass burning component. The simulations were initialized with and without the biomass burning component using the background levels as the reference condition. The results from the simulations were then compared to ground-based observations. Studies investigating the contribution of stratospheric ozone to the tropospheric ozone budget have shown that, during the biomass burning season, the contribution of stratospheric ozone is insignificant in the lower troposphere over Irene [53,54].

## 3. Results and Discussion

In this section, the aerosol, tropospheric ozone and UVI patterns are presented. The relationship between aerosols/tropospheric ozone and UVI are presented to demonstrate the radiative effect of aerosols and tropospheric ozone.

### 3.1. Aerosol Climatology

Daily averages of AOD and AE from 2011 to 2018 were obtained for the AERONET station at the CSIR in Pretoria. The aerosol observations at the CSIR showed that AOD (Figure 2) reached an annual minimum in June (0.23) and increased to an annual maximum in September (0.46).

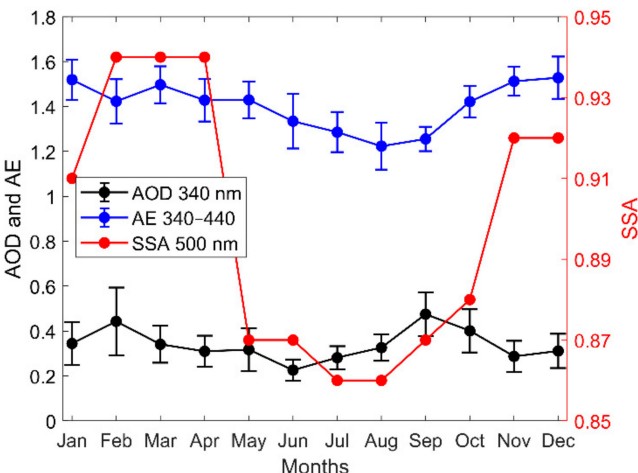

**Figure 2.** Monthly means and standard deviations of aerosol properties for 2011 to 2018 from sun photometer observations at the CSIR: Aerosol optical depth (AOD) at 340 nm (black) and Ångström Exponent (AE) (blue) in the 340–440 nm spectral band as well as monthly mean single scattering albedo (SSA) at 550 nm (red) from the MACv2 aerosol climatology.

The two annual AOD maxima (in February and September) coincided with summer (December, January, and February) and spring (September, October, and November), respectively. These peaks have also been identified in previous studies [21,55]. During spring, emissions from biomass burning could increase AOD, while in summer months, strong convection could result in the disturbance of surface particles thereby increasing AOD [56].

The AE parameter was inversely related to aerosol particle size. AE values (Figure 2) had a small range and were dominated by fine aerosols from anthropogenic emissions [21]. During the biomass burning season, AE values indicated that coarser particles were present which were typically associated with aerosols from biomass burning. AE increased from a minimum in August (~1.25) to a maximum in December (~1.55). Figure 2 also shows an inverse relationship between the seasonal variations of AOD and AE, which highlights the presence of biomass burning and urban/industrial aerosol types over Pretoria. The monthly SSA averages from the MACv2 aerosol climatology were lowest during the

austral winter and highest during summer. A similar seasonal cycle in SSA has been described in a previous study over Pretoria [21].

### 3.2. Tropospheric Ozone

Using the ozone sounding data from the Irene weather station, the tropospheric ozone profile and tropospheric ozone total column were derived for the periods 1998 to 2018 (369 ozonesondes) (Figure 3a,b) and 2012 to 2018 (113 ozonesondes) (Figure 4a,b). For the period 1998 to 2018, the tropospheric ozone total column (Figure 3a) reached an annual maximum (45 DU) around October each year. This was followed by a decrease over the summer months to an annual minimum in May, with the largest variability occurring during the spring and summer months. The vertical tropospheric ozone profile over Irene (Figure 3b) showed that the highest ozone mixing ratio took place during spring (from August to October) and extended down to 3.5 km. The vertical ozone profile was similar to that reported by Sivakumar et al. [20] with the increases in tropospheric ozone occurring during the biomass burning season.

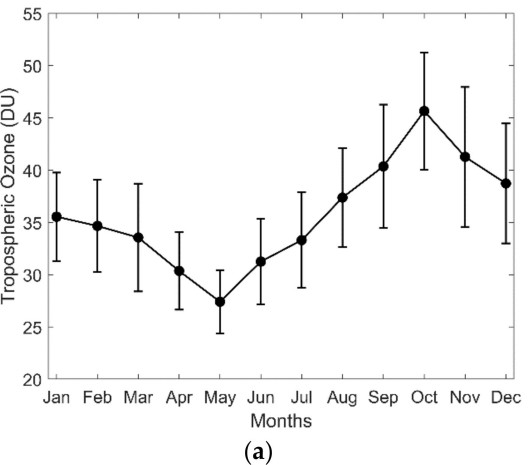

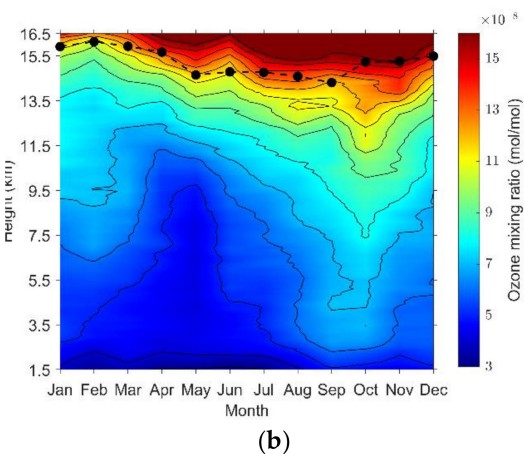

(**a**)         (**b**)

**Figure 3.** Tropospheric ozone data obtained from ozonesondes launched at Irene: (**a**) Monthly mean and standard deviation of the total tropospheric ozone column obtained from ozonesondes between 1998 and 2018; (**b**) Monthly mean ozone mixing ratio from 1.5 to 16.5 km above sea level and monthly mean lapse-rate tropopause obtained from ozonesondes between 1998 and 2018 (dashed black line).

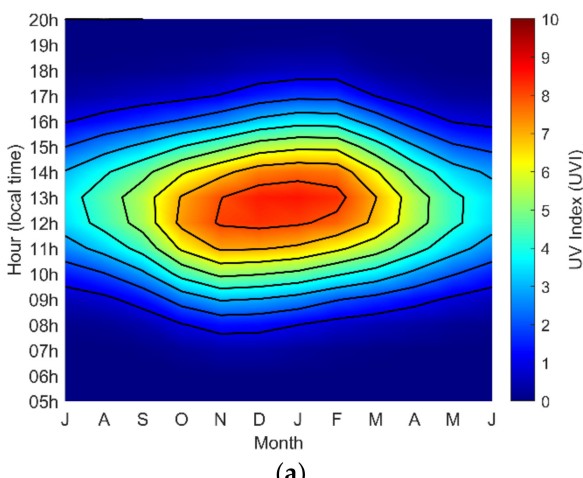

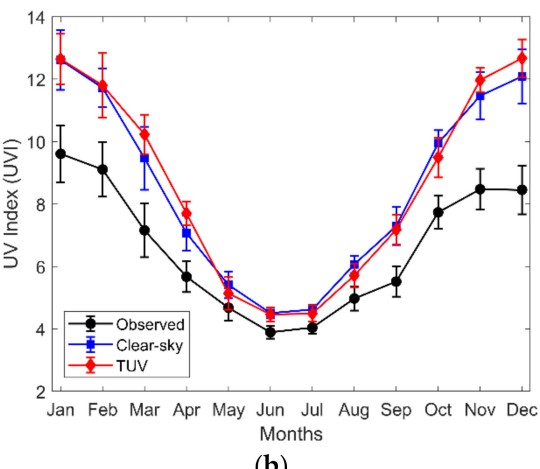

(**a**)         (**b**)

**Figure 4.** Solar Ultraviolet Index (UVI) observations for Bolepi House, Pretoria from 2009 to 2018. (**a**) Monthly and hourly averages of UVI for July to June; and (**b**) UVI at solar noon for observed all-sky and clear-sky UVI compared to modelled clear-sky UVI for January to December (red line).

The seasonal peak in tropospheric ozone (Figure 3a) is in agreement with previously published work [9,12,20] and occurred in the austral spring when STE enhanced tropospheric ozone concentrations due to the Brewer–Dobson circulation and coincided with the peak of the biomass burning season [57]. Tropospheric ozone formation favors drier conditions that occur near the end of the dry season which further contribute to the seasonal peak in tropospheric ozone [18,58]. Although the main biomass burning region is to the north and east of Irene, the anti-cyclonic air mass from the biomass burning region as well as the emission of ozone precursors from coal-fired power stations affect tropospheric ozone at Irene [12,58].

### 3.3. Observed and Modelled UVI Levels

Hourly solar UVB radiation data were used to determine monthly and hourly averages of UVI. The observed clear-sky UVI levels were compared to modelled clear-sky UVI levels. The hourly and monthly averages (Figure 4a) showed that UVI is at a maximum during the summer months and highest daily between 12:00 and 14:00 local time. This was further demonstrated by the solar noon UVI values for all-sky and clear-sky conditions (Figure 4b). The higher observed clear-sky solar noon UVI values—with respect to the all-sky UVI values—indicated that the clear-sky determination method was able to remove the effect of clouds on UVI.

The monthly averaged relative difference between the TUV model and observed clear-sky UVI at solar noon ranged between −4% and 29% with an annual relative difference of 7%. Between November and April, the modelled clear-sky values were higher than the observed clear-sky UVI. The monthly-averaged relative difference between the TUV model and observed all-sky UVI ranged between 18% and 87%. The largest difference between the modelled and observed UVI occurred during summer months and may be related to the increase in cloud cover associated with convection during summer months due to synoptic-scale circulation.

Comparisons with observed and modelled UVI in the Southern Hemisphere showed a small relative difference of approximately 10% on clear-sky days [59,60]. To our knowledge, this is the first study in South Africa to compare modelled and observed surface UVI and investigates the effect of atmospheric parameters on surface UVR over South Africa. The TUV model simulations may vary from the observed surface UVI due to factors such as the vertical distribution of aerosols, using AE for 340–440 nm and the climatological ozone and temperature profiles derived by McPeters and Labouw [49] that were used in the simulations. Future research could use satellite lidar observations from the Cloud-Aerosol Lidar with Orthogonal Polarization (CALIOP) instrument to investigate the vertical distribution of aerosols. Furthermore, an ozone and temperature profile specific to Irene could be utilized by using data from ozonesondes and satellites.

### 3.4. Anomalous AOD over Pretoria

Between August 2017 and October 2017, 31% of AOD observations from the AERONET station at the CSIR were one-standard-deviation above the respective monthly average. The AOD and UVI observations were compared during this period (i.e., August to October 2017) to determine if the anomalously high AOD resulted in lower surface UVR. AOD had seasonal peaks in February and September (Figure 2). Between August and October 2017 (Figure 5a), 24 days were one-standard-deviation above the monthly AOD average. Between 2 September and 2 October 2017, the majority of AOD observations were well above one-standard-deviation from the monthly mean. Between 3 October and 18 October 2017, AOD was below one-standard-deviation from the monthly average. Figure 5b shows the daily and monthly averages of the modelled clear-sky UVI. Between 2 September and 2 October 2017 when AOD was anomalously high, UVI was below one standard deviation from the monthly mean. In October 2017, when AOD was below the monthly average, the corresponding UVI values were above the monthly average.

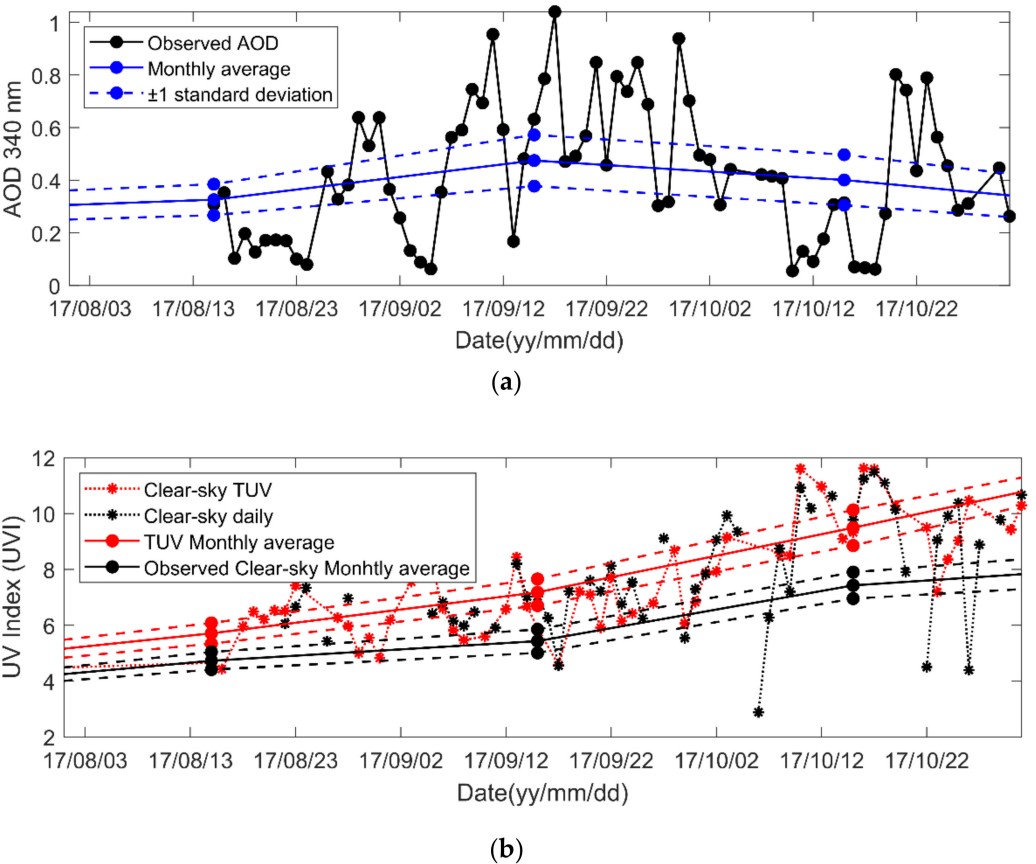

**Figure 5.** (**a**) Daily AOD values at 340 nm from August to October 2017 as recorded by the AERONET station at the CSIR and the monthly AOD averages and standard deviations (during the 2011–2018 period); (**b**) Daily modelled clear-sky UVI and modelled and observed clear-sky monthly averages and standard deviations from August to October 2017.

From August to October 2017, higher AOD observations occurred simultaneously to lower UVI values and vice versa. This inverse relationship between AOD and UVI—with anomalously high AOD values—has been observed in previous studies [61,62]. Using dispersion modelling, future research may trace the origin of specific biomass burning episodes and investigate the radiative effect over Pretoria.

*3.5. Effect of Aerosols and Tropospheric Ozone*

To assess the sensitivity of surface solar UVR to aerosols and tropospheric ozone, the TUV model was used to simulate surface solar UVR in four different scenarios and modelled UVI was compared to ground-based observations. The simulations were initialized for the period August to October (2011–2017) with and without the biomass burning component of aerosols and tropospheric ozone. The biomass burning component used the monthly averages of tropospheric ozone and aerosols optical properties (AOD, AE, and SSA) between August and October, while the background level used the monthly averages of tropospheric ozone and aerosol optical properties between November and July as described in Section 2.2.

Table 1 presents the relative difference (%) between the modelled and observed clear-sky UVI in each of the simulations over the three months. The aerosol and tropospheric ozone columns in Table 1 indicate whether the background level or biomass burning emissions were used in the respective simulations. The first simulation (Simulation 1) was a reference simulation and included the aerosol and tropospheric ozone levels between August and October as seen in Figures 2 and 3a. In this simulation, the modelled clear-sky UVI was 3% than observed clear-sky UVI during September months. In Simulation 2, when aerosols and tropospheric ozone from biomass burning were not included, the modelled

clear-sky UVI was 11% higher in September months compared to the observed clear-sky UVI; a change of approximately 14% compared to the reference simulation. In Simulation 3, where tropospheric ozone from biomass burning was excluded and the background level was used, the modelled clear-sky UVI was 2% lower than the observed which was similar to Simulation 1. In Simulation 4, where aerosols from biomass burning were excluded, the modelled UVI was 10% higher than the observed UVI and was similar to Simulation 2 where both aerosols and tropospheric ozone from biomass burning were excluded. Simulations 2, 3, and 4 indicated that increases in aerosols and tropospheric ozone contributed to the reduction in UVR flux reaching the surface and that aerosols from biomass burning have a larger radiative effect.

**Table 1.** Relative differences (%) between modelled and observed UVI from Tropospheric Ultraviolet-Visible (TUV) model simulations with and without aerosols and tropospheric ozone for August to October (2011 to 2017) and the average relative difference between August and October (ASO).

|  | Aerosol | Tropospheric ozone | RD—August | RD—September | RD—October |
|---|---|---|---|---|---|
| Simulation 1 | BB * | BB | −4 | −3 | −7 |
| Simulation 2 | BGL ** | BGL | −2 | 11 | 2 |
| Simulation 3 | BB | BGL | −4 | −2 | −6 |
| Simulation 4 | BGL | BB | −2 | 10 | 1 |

* BB—biomass burning, ** BGL—background level.

Although the changes in Simulation 3 were small compared to the reference simulation, increases in tropospheric ozone may result in decreases in UVR at the surface due to Rayleigh scattering of denser air in the lower levels as shown by other studies [63,64]. Positive decadal trends in tropospheric ozone have been identified over Pretoria [65,66] and these trends may further contribute to the radiative effect of tropospheric ozone.

Future research should investigate the radiative impact of nitrogen dioxide as well as tropospheric aerosols and tropospheric ozone on UVI levels at the surface. Furthermore, research should be done at all SZAs and not be limited to solar noon. Although the three stations (i.e., CSIR, Irene and Bolepi House) were located relatively close to each other, factors such as cloud cover and albedo may have differed between the stations and this could have influenced the comparison between observed and modelled UVI data.

## 4. Conclusions

The aim of this study was to investigate the effect of aerosols and tropospheric ozone on surface UVR over Pretoria during the biomass burning season. The study modelled surface UVR with and without aerosols and tropospheric ozone from biomass burning and compared the modelled UVR to surface observations. The aerosol loadings and atmospheric composition were affected by biomass burning which reached a peak near the end of the dry winter season. Over Pretoria, AOD reached a maximum value of 0.46 in September and tropospheric ozone reached a maximum of 45 DU in October. The study area was predominantly affected by fine aerosol particles which increased in size during the biomass burning when AOD was at a maximum.

A comparison between modelled and observed clear-sky UVI levels at solar noon showed a small relative difference of 7% on clear-sky days. Between August and October 2017, anomalously high AOD levels were observed over Pretoria. Investigation of this event showed that higher AOD values corresponded with lower UVR levels. In the TUV model simulation—which excluded aerosols and tropospheric ozone from biomass burning—a change of 9% relative to the reference simulation was observed. Furthermore, in the simulation that excluded aerosols from biomass burning there was a relative difference similar to the simulation that excluded both aerosols and tropospheric ozone. This demonstrated that the radiative effect of aerosols was larger than the radiative effect of tropospheric ozone. Future research on the radiative effect of aerosols and trace gases,

particularly in regions that are affected by emissions from biomass burning, is considered important.

**Author Contributions:** Conceptualization, D.J.d.P., C.Y.W., H.B., K.L., and T.P.; methodology, D.J.d.P., C.Y.W., H.B., and T.P. formal analysis, D.J.d.P.; writing—Original draft preparation, D.J.d.P., C.Y.W., H.B., and T.P.; writing—Review and editing, D.J.d.P., C.Y.W., H.B., K.L., and T.P. All authors have read and agreed to the published version of the manuscript.

**Funding:** D.J.d.P. received a doctoral scholarship from the University of Pretoria and a scholarship from the French Embassy in South Africa. C.Y.W. receives funding from the South African Medical Research Council, the National Research Foundation of South Africa, and the University of Pretoria. The APC was funded by LACy (Laboratoire de l'Atmosphère et des Cyclones).

**Institutional Review Board Statement:** Not applicable.

**Informed Consent Statement:** Not applicable.

**Data Availability Statement:** The datasets used in this study are freely available from the relevant sources.

**Acknowledgments:** The authors would like to thank the South African Weather Service for providing UVB and SHADOZ network for ozone sounding data. Furthermore, we would like to thank AERONET for the aerosol data. Authors acknowledge the French Embassy in Pretoria, the South-African PROTEA programme and the CNRS-NRF International Research Project ARSAIO (Atmospheric Research in Southern Africa and Indian Ocean), for supporting research activities.

**Conflicts of Interest:** The authors declare no conflict of interest.

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
