# Peer review of "Solar Ultraviolet Radiation in Pretoria and Its Relations to Aerosols and Tropospheric Ozone during the Biomass Burning Season"

_atmosphere, doi:10.3390/atmos12020132_

Round 1

Reviewer 1 Report

The paper has still major problems and appears to be premature and not state of the art. The abstract promises results that are not provided in the later sections since the authors ignore the comments to that in the first review concerning the separation of the effects of biomass burning. The description of
the model setup still lacks essential things but now reveals the use of an inappropriate input data set. It is a 'no-go' just to remove observations out of an essential figure instead of the suggested improvement of the analysis.
There are also still plenty of grammar issues.
Nevertheless the paper contains some valuable data and an interesting list of references.

Details:
Abstract: Effects of biomass burning are mentioned but the simulations appear to show only the effects of total tropospheric ozone including the one transported downward from the stratosphere (like confirmed by the response to the first review). Also the aerosol is not separated (see table 1).

Line 36: Increase and decrease messed up.

Line 73: Important parts are missing like the separation of biomass burning effects.

Eqn 1: A minus sign is missing. Better use the definition of Wikipedia since AE is derived from for 2 discrete wavelengths later.

Line 103: A short description is still missing since the reference is difficult to access.

Line 106: This is UV-A. Often you cannot use this Angstrom coefficient to extrapolate to UV-B (e.g. in case of mineral dust). At least this introduces an
uncertainty.

Line 126: Put a typical altitude into parentheses.

Eqn 2: Unit is still wrong, UVI has not the unit 1/h! (remove that at MED?).

Line 162: This is now partially missing in figures.

Line 173ff: Leave out NO2 since it is not used. Are O3 data checked for consistency concerning the vertical integral (total ozone) and adjusted?
Are temperature profile in the troposphere from ozone sondes (not available from MLS)? More details please.

Line 180: I suppose you mean inversion? Please describe. See line 103.

Line 184f: Skip SO2 here or use proper data, this is distracting.

Line 186f: This is not state of the art. There are also data from measurement campaigns in the neighborhood (e.g. ACTRIS, ATOM), from the AEROCOM community, and in "Aerosols and their climatic effects, edited by: Gerber, H. and Deepak, A., pp. 133–177, A. Deepak, Hampton, VA, 1984". Is the profile scaled with the observations of AOD? More details please. Elterman is not standard but special and outdated. The data are strongly perturbed by a major volcanic eruption, causing a significant AOD in the stratosphere, and based only on one station at about 35N in the US Rockies. The tables do not include a distinct planetary boundary layer.

Line 187: Something missing.

Line 205: 'Climatology' is not the best word here.

Line 225: Large particles from biomass burning and small from urban pollution? Is there a dust contribution?

Line 234ff: Separate in correct way, only the lower part is related to biomass burning (up to 6km), the rest to STE. Improve language! Please provide O3 down to the surface. What is height here? Above surface at station or above sea level?

Figure 3: Provide the same for the subset used in the UV calculations (2 more frames). Is there a trend related to filters in stacks of power plants or more
biomass burning?

Line 259: This is in contradiction to the figure, please correct number.

Line 265: Explain what has been done in the years without data.

Line 270ff: It is not enough to state that a constant SSA is not appropriate. Better repeat calculations in a more sophisticated way.

Line 281: Time span?

Line 293ff: Caption inconsistent to figure (part b). It is bad practice just to leave out the observations to avoid a correct analysis concerning cloud
effects and model artifacts due to oversimplifications.

Line 315ff: What is used here for tropospheric ozone? Zero (nonsense) or the profile of the month with the minimum value or what? Please clarify. Same for aerosol, based on minimum AOD or what? You should try to separate the seasonal cycle due to BB from that due to STE. This includes to adjust the vertical integrals for ozone and aerosol extinctions.

Table 1: The sensitivity studies do not provide what is promised in the abstract. The numbers are dominated by the natural background and therefore too large. In the troposphere is always ozone from STE. Setting this to zero is artificial. Biomass burning and pollution effects cannot be analysed this way.
You need some (seasonally dependent) background climatology, also for aerosol. Or focus on the region below 6km with explanation.
The studies have to be repeated in a more useful way.

Line 355: This is misleading, see comment to Table 1.

Line 467: Which year, when accessed?

Line 505: Which year? URL? Publisher?

Line 512: Provide more details, at least AFCRL-68-0153. You should read the report in detail to see that the data are not appropriate.

Remove html-commands!

Author Response

Revision and response to Reviewers comments for the manuscript entitled “The effects of tropospheric aerosols and ozone on solar ultraviolet radiation over Pretoria, South Africa”

Reviewer 1 (R1)

The paper has still major problems and appears to be premature and not state of the art. The abstract promises results that are not provided in the later sections since the authors ignore the comments to that in the first review concerning the separation of the effects of biomass burning. The description of the model setup still lacks essential things but now reveals the use of an inappropriate input data set. It is a 'no-go' just to remove observations out of an essential figure instead of the suggested improvement of the analysis. There are also still plenty of grammar issues. Nevertheless, the paper contains some valuable data and an interesting list of references.

Response: We thank the Reviewer for all of the comments that have helped us to improve the manuscript. We have done our best to address all the Reviewers’ comments.

Details:

Abstract:

  1. Effects of biomass burning are mentioned but the simulations appear to show only the effects of total tropospheric ozone including the one transported downward from the stratosphere (like confirmed by the response to the first review). Also, the aerosol is not separated (see table 1).

Response: The abstract has been revised to include the revision with regard to the model simulations as indicated in Response 24 (R1).

  1. Line 36: Increase and decrease messed up.

Response: The sentence has been revised for clarity at line 36 “UVR is classified into three bands: UVA (315-400 nm), UVB (280-315 nm) and UVC (100-280) where the absorption of UVR by stratospheric ozone increases within the UVB spectrum.”

  1. Line 73: Important parts are missing like the separation of biomass burning effects.

Response: The separation of aerosols and tropospheric ozone from biomass burning have been included at line 72 and further revisions have been made throughout the manuscript.

  1. Eqn 1: A minus sign is missing. Better use the definition of Wikipedia since AE is derived from for 2 discrete wavelengths later.

Response: Equation 1 has been corrected in the revised manuscript as suggested.

  1. Line 103: A short description is still missing since the reference is difficult to access.

Response: Further detail has been given on the inversion algorithm in line 104 “The AERONET inversion algorithm is used to provide aerosol optical properties such as single scattering albedo (SSA) which is derived from direct and diffuse radiation measurements from sun photometers [29].  The inversion algorithm assumes that the vertical distribution of the aerosols is similar to global models, particles are partitioned into spherical and non-spherical and it accounts for the gaseous absorption by ozone, nitrogen dioxide and water vapour. Furthermore, the algorithm provides information on the quality of the output parameters produced [30].”

  1. Line 106: This is UV-A. Often you cannot use this Angstrom coefficient to extrapolate to UV-B (e.g., in case of mineral dust). At least this introduces an uncertainty.

Response: The Pretoria AERONET station provides AE data at 440-870 nm, 380-500 nm, 440-675 nm, 500-870 nm and 340-440 nm. We have included using AE for 340-440 nm as a limitation at line 277 “Furthermore, variations in the vertical aerosol distribution and using AE for 340-440 nm used in the TUV model could lead to specific modelling errors.”

  1. Line 126: Put a typical altitude into parentheses.

Response: An approximate burst altitude of the balloon has been included.

  1. Eqn 2: Unit is still wrong, UVI has not the unit 1/h! (remove that at MED?).

Response: The equation has been corrected as suggested.

  1. Line 162: This is now partially missing in figures.

Response: This has been corrected as indicated in Response 23 (R1)

  1. Line 173ff: Leave out NO2 since it is not used. Are O3 data checked for consistency concerning the vertical integral (total ozone) and adjusted?
    Are temperature profile in the troposphere from ozone sondes (not available from MLS)? More details please.

Response: In the TUV model simulations, NO2 was used in the input data for the model and therefore we have included it. From line 245, further details on the ozone and temperature profiles used have been provided. Furthermore, in line 405 we have mentioned using a climatological profile as a limitation.

  1. Line 180: I suppose you mean inversion? Please describe. See line 103.

Response: The correction has been made at line 183. Please refer to Response 5 (R1) relating to the inversion algorithm.

  1. Line 184f: Skip SO2 here or use proper data, this is distracting.

Response: The mention of SO2 data has been removed.    

  1. Line 186f: This is not state of the art. There are also data from measurement campaigns in the neighborhood (e.g., ACTRIS, ATOM), from the AEROCOM community, and in "Aerosols and their climatic effects, edited by: Gerber, H. and Deepak, A., pp. 133–177, A. Deepak, Hampton, VA, 1984". Is the profile scaled with the observations of AOD? More details please. Elterman is not standard but special and outdated. The data are strongly perturbed by a major volcanic eruption, causing a significant AOD in the stratosphere, and based only on one station at about 35N in the US Rockies. The tables do not include a distinct planetary boundary layer.

Response: We agree with the Reviewer that the Elterman extinction profile is most probably unsuitable, but we currently only have the integrated aerosol optical depth measurements. For future projects we plan to investigate the extinction profile using observations from CALIPSO. We have noted this as a limitation in the manuscript at line 405.

  1. Line 187: Something missing.

Response: This has been corrected in the revised manuscript at line 191 “Since AOD and AE data were only available for 1 July 2011 to 31 May 2018, the TUV model was used to calculate the clear-sky UVI levels for this period.”

  1. Line 205: 'Climatology' is not the best word here.

Response: The word climatology has been replaced with pattern / trend in the manuscript.

  1. Line 225: Large particles from biomass burning and small from urban pollution? Is there a dust contribution?

Response: Additional information has been added in line 227 “During the biomass burning season, AE values indicated that coarser particles are present which are typically associated with aerosols from biomass burning.” Furthermore, previous studies have shown that dust contributes to the aerosol loading when convection occurs, as stated in line 218.

  1. Line 234ff: Separate in correct way, only the lower part is related to biomass burning (up to 6km), the rest to STE. Improve language! Please provide O3 down to the surface. What is height here? Above surface at station or above sea level?

Response: The figure captions have been checked and the language has been revised as in Response 18 (R1).

  1. Figure 3: Provide the same for the subset used in the UV calculations (2 more frames). Is there a trend related to filters in stacks of power plants or more biomass burning?

Response: The figures have been added as suggested and the discussion has been adapted accordingly.

  1. Line 259: This is in contradiction to the figure, please correct number.

Response: The correction has been made in the revised manuscript at line 277.

  1. Line 265: Explain what has been done in the years without data.

Response: Only days with a complete observation recorded were considered for analysis as stated in line 166 “From the hourly data, the monthly hourly averages, and standard deviations of UVI at solar noon were calculated only using days with a complete observation record.”

  1. Line 270ff: It is not enough to state that a constant SSA is not appropriate. Better repeat calculations in a more sophisticated way.

Response: The TUV model simulations have been revised using data from an aerosol climatology as stated in line 185 “The inversion algorithm resulted in few SSA data points and as a result, monthly averages of SSA (550 nm) were obtained for an area over Irene from the MACv2 aerosol climatology [50].”

  1. Line 281: Time span?

Response: Time span has been added.

  1. Line 293ff: Caption inconsistent to figure (part b). It is bad practice just to leave out the observations to avoid a correct analysis concerning cloud effects and model artifacts due to oversimplifications.

Response: The figure and caption have been revised as suggested.

  1. Line 315ff: What is used here for tropospheric ozone? Zero (nonsense) or the profile of the month with the minimum value or what? Please clarify. Same for aerosol, based on minimum AOD or what? You should try to separate the seasonal cycle due to BB from that due to STE. This includes to adjust the vertical integrals for ozone and aerosol extinctions.

Response:  The biomass burning and seasonal AOD and tropospheric ozone have been separated. The method has been described in Section 2.5 and the results in Section 3.5 have been revised as well.

To address STE, we have included the following in line 210 “Studies investigating the contribution of stratospheric ozone to the tropospheric ozone budget have shown that during the biomass burning season, the contribution of stratospheric ozone is insignificant over Irene [52,53].”

  1. Table 1: The sensitivity studies do not provide what is promised in the abstract. The numbers are dominated by the natural background and therefore too large. In the troposphere is always ozone from STE. Setting this to zero is artificial. Biomass burning and pollution effects cannot be analysed this way. You need some (seasonally dependent) background climatology, also for aerosol. Or focus on the region below 6km with explanation. The studies have to be repeated in a more useful way.

Response: The method and results have been revised as indicated in Response 24 (R1).

  1. Line 355: This is misleading, see comment to Table 1.

Response: The method and results have been revised as indicated in Response 24 (R1).

  1. Line 467: Which year, when accessed?

Response: We have amended the article title, also ‘no date’ since there is no date of publication, and the access date which was the 7 March 2020.

  1. Line 505: Which year? URL? Publisher?

Response: The date 22-23 September 2019, URL and access date and the Publisher have been added.                  

  1. Line 512: Provide more details, at least AFCRL-68-0153. You should read the report in detail to see that the data are not appropriate.

Response: Further details and limitations have been provided in the manuscript and have been detailed in Response 24 (R1).

  1. Remove html-commands!

Response: Html commands were removed.

Reviewer 2 Report

An extensive editing of English language and style is required. The text has been improved in some parts but some discussions and methods used in this study are still not adequately explained and the description needs to be improved. Some of the statements are incorrect and need to be reformulated. More accuracy is needed both in the exposition and in some of the scientific analyses of the study.The manuscript is generally not well written, extensive English checking is strongly recommended.

Please refer to the attached docx for the specific comments.

Author Response

Reviewer 2 (R2)

The manuscript "The effects of tropospheric aerosols and ozone on solar ultraviolet radiation over Pretoria, South Africa” by David Jean du Preez presents a study that is aimed to evaluate the impact of tropospheric aerosols loading and tropospheric ozone concentration on solar UVR over Pretoria during the biomass burning season, using ground measurements and modelled data. An extensive editing of English language and style is required. The text has been improved in some parts, but some discussions and methods used in this study are still not adequately explained and the description needs to be improved. Some of the statements are incorrect and need to be reformulated. More accuracy is needed both in the exposition and in some of the scientific analyses of the study. The manuscript is generally not well written, extensive English checking is strongly recommended.

Response: We have revised the manuscript according to all the Reviewers’ comments and we have had the manuscript reviewed for English grammar.

Major comments:

Introduction:

  1. Line 40: “Atmospheric aerosols have a direct or indirect effect on…”→ direct and indirect

Response:  The sentence has been revised at line 40 “Atmospheric aerosols have a direct and indirect effect”

  1. Lines 72-75: “The results from the data analysis show the seasonal cycle of aerosol, tropospheric ozone and UVR are…” in this sentence something is missing, please rephrase.

Response: The sentence has been revised at line 72 “The results from the data analysis show the seasonal cycle of aerosol, tropospheric ozone and UVR as well as a case study and model simulations to demonstrate the radiative effect of aerosols and tropospheric ozone.”

Data and methods:

  1. Line 95 “Aerosol Optical Depth (AOD) observations are made…..and is an indication…” please rephrase

Response: The sentence has been revised at line 96 “Aerosol optical Depth (AOD) is an indication of the distribution of aerosols in a column of air. Observations at 935 nm are used to estimate columnar water vapour.”

  1. Line 97 “The eighth spectral band is used to estimate columnar water vapour.” Please check the wavelength (1640?) and add a reference for this.

Response: The spectral band description has been revised in Response 3 (R2). The wavelength of 1640 nm is correct and can be found on the AERONET site. The reference has been added at line 97.

  1. Line 102: in equation 1, the τα should be τλ

Response: Equation 1 has been revised.

  1. Line 103: “The inversion algorithm.” What inversion algorithm? Please rephrase and clarify.

Response: Further detail has been given on the inversion algorithm in line 104 “The AERONET inversion algorithm is used to provide aerosol optical properties such as single scattering albedo (SSA) which is derived from direct and diffuse radiation measurements from sun photometers [29].  The inversion algorithm assumes that the vertical distribution of the aerosols is similar to global models, particles are partitioned into spherical and non-spherical and it accounts for the gaseous absorption by ozone, nitrogen dioxide and water vapour. Furthermore, the algorithm provides information on the quality of the output parameters produced [30].”

  1. Lines 104-106: “The SSA are important” please correct and rephrase

Response: The sentence has been revised at line 109 “SSA is an important factor related to the radiative effect of aerosols and represents the ratio between the scattering and extinction efficiencies of aerosols [31].”

  1. Line 106: You cite AE derived from AERONET for 340-440 nm. Looking to Pretoria AERONET site AE is provided for 440-870 nm. Please explain if you calculated the AE as described (using AOD values from AERONET, I imagine) or If you used AERONET data of AE directly (in this case correct the range)

Response: The AE data for 340-440 nm was obtained directly from the AE website as stated in line 112 “The level two daily average data of AOD (340 nm) and AE (340 – 440 nm) were obtained from the AERONET website (aeronet.gsfc.nasa.gov/) for the period from 1 July 2011 and 31 May 2018”. Additionally, the Pretoria AERONET station provides AE data at 440-870 nm, 380-500 nm, 440-675 nm, 500-870 nm and 340-440 nm.

  1. Line 115-116 Please avoid expressions like “there were” or “there was”. Moreover, ozonesondes are not conducted (ozone soundings are conducted). Please correct and rephrase.

Response: We have amended the use of ‘there was/were’ and we have changed ‘conducted’ to ‘launched’.

  1. Line 117-118 “The solution of potassium iodide (KI) and KI….” Something is wrong here: please check.

Response: The sentence has been corrected “The solution of buffered potassium iodide (KI) and saturated solution of KI was used in the cathode and anode cell, respectively.”

  1. Lines 117-122: All the description is very confused.

Response: The text has been revised: Between 1998 and 2018, 369 ozonesondes were launched (however no ozonesondes were launched between 2008 and 2012). The ozone soundings were conducted twice a month using an electrochemical cell comprising a cathode and anode cell. The solution of buffered potassium iodide (KI) and saturated solution of KI were used in the cathode and anode cell, respectively. An ion bridge connected the cells which allows for electrons to flow between the two chambers and ozone was measured using the iodine/iodide electrode reactions.

  1. Line 128: “than 2 K.km-1 for”. Eliminate the point between K and km

Response: Point removed.

  1. Line 129: “average tropospheric column and average profile” → average ozone tropospheric column and average ozone profile

Response: The word ‘ozone’ has been included.

Observe UVB data from Pretoria

  1. Line 132: “The SAWS …” Please add a reference

Response: The data was obtained directly form the SAWS. A reference for the dataset has been added.

  1. Line 135-142 This description is very confused, please

Response: The sentences have been rewritten: The instrument provided an analogue voltage output proportional to the measured radiation [33] that was given in Minimal Erythemal Dose (MED) units. One MED is approximately 210 Jm-2, where MED is a metric used to express the minimal erythemal dose required to induce erythema (also known as sunburn) [34].         

  1. Line 150 “..were determined..” Line 153 “is determined”. Please use the same verbal tense in the manuscript.

Response: The manuscript has been reviewed to ensure past tense is used throughout the text.

  1. Line 154 “for monotonic increases and decreases before and after solar noon” Please complete: increases and decreases of what?

Response: ‘in UVI values’ has been added.

  1. Line 156 “the monthly and hourly averages” What kind of averages did you determine? It is not clear. You said that your instrument provides measurements at hourly intervals. Did you mean monthly hourly averages? (i.e., monthly averages given at each hour).

Response: The suggested correction has been made in the manuscript in line 166.

  1. Line 162 “modelled data” Please add a reference to the next Section

Response: A reference has been added to the section.

Modelled UVR over Pretoria

  1. Lines from 164 to 166: Please rephrase

Response: Sentences fixed and rephrased as recommended.      

  1. Line 169 radiance, irradiance

Response: Amended to irradiance.

  1. Line 183 Please correct the verb

Response: Amended to ‘have’.

  1. Line 187-188 Rephrase

Response: Conjunction has been added.

Results and discussion

  1. Lines 211-214 Something is probably missing in the sentence, please check and correct.

Response: The conjunction ‘and’ has been included.

  1. Line 222-223: 1.25 and 1.55 is not what the figure shows. Please add the exact value or write “nearly” 1.25 and 1.55

Response: The symbol for approximately ~ has been added.

Tropospheric ozone

  1. Line 228: “tropospheric ozone total column” please correct in both cases here.

Response: Amended as requested.

  1. Line 233: In what sense the vertical ozone profile is similar?

Response: The revision has been included in the manuscript at line 240.

Observed and modelled UVI levels

  1. Line 256 ”relative to” → “with respect to”

Response: Amended as requested.

  1. Line 257 “is able”

Response: Amended as requested.          

Anomalous AOD over Pretoria

  1. Line 286 Remove the sentence “In August…”

Response: Sentence has been removed.

  1. Line 295: remove “observed data”

Response: Text was removed.

  1. Line 299: Please correct the sentence.

Response: Sentence has been refined.

Effect of aerosols and tropospheric ozone

  1. Line 303: correct “tropospheric”

Response: Subheading title has been corrected.

  1. Line 318-319 “As for simulation 3, it does not consider the seasonal loading of biomass-burning aerosols over the study site. “ Please rewrite the sentence.

Response: Sentence has been refined.

  1. Line 319-320 “ is more than doubled” Line 321 Please correct the verbal tense.

Response: Tense corrected.

  1. Line 324 “in September”. Please remove “months”

Response: Month removed.

  1. Line 329-332 This sentence has white spaces, and the punctuation is absent. Please check and correct.

Response: Space have been removed and punctuation has been fixed.  

  1. Line 332-335 Please clarify that these are results obtained by other studies

Response: ‘as shown by other studies’ has been added.

  1. Line 336-340 Please rewrite these sentences. English check is needed.

Response: Sentences have been corrected.

Conclusion

  1. In the conclusion the authors should resume the aim of the study, the methods used, and the results obtained. Here the authors introduced new information (e.g., lines 284-286) that should be introduced and discussed in the paper in the introduction paragraph, as already mentioned in my first review.

Response: The conclusion has been revised and special attention has been given to ensure that new information has not been included.

Reviewer 3 Report

The authors have provided a thoroughly revised manuscript and I recommend publication in its current form.

Author Response

Reviewer 3 (R3)

  1. The authors have provided a thoroughly revised manuscript and I recommend publication in its current form.

Response:  We would like to thank the Reviewer for reviewing this manuscript.

Round 2

Reviewer 1 Report

The paper is better now in a lot of points but still has flaws which have to be corrected to allow publication.

Title: "relations to".
Abstract: Line 26: provide numbers from table 1 (aerosol up to 13% and ozone up to 1%).
Keywords: add "ozone; UV" (TUV is too special).
Eqn 2, line 159: please correct: numerator: times MED[h^-1], denominator: 3600[s/h] (how it was written it is nonsense, there was a misunderstanding).
Line 204ff: zonal average in the stratosphere or down to surface? What is done in the troposphere? Are the average sonde data used? This is not clear from line 197. That they are not used appears too late in the text (line 368).
Line 237: this holds for the lower troposphere only.
Line 243: better skip the word "trends" here (in the subtitle "climatology" was better).
Line 254: "maxima"
Figure 3c and d and text lines 283ff: there must be something wrong, at least in January to March. It cannot be that lapse rate tropopause (same as in part b in contrast to caption?) and ozone tropopause differ so much. Check how many data points are there in each month. Are there bad profiles? If you cannot solve this, better skip this part since it appears to be not used later, maybe keep a remark in the text.
Line 326: Not 11 to 13h? Or is the time zone off from true solar local time? [just a comment]
Line 363: skip "."
Figure 5: Don't compare apples and oranges. Leave out the lines with average observed all-sky data (?, not indicated in caption!) and use the clear-sky subset for observed averages as shown in Figure 4 but for the time period in the caption to support what is written in the text. Part a and b should have the same size.
Table 1: "(%) between modelled"? You may skip the last column and focus on September in the text. Footnotes are twice!
Line 439: typo
References:
Line 557: typo
Line 576: URL?
Line 589: An organization has no first name.
Line 590: Is there an URL?
Line 622: Reference incomplete, provide at least "AFCRL-68-0153" to enable the reader to find this technical report.

Author Response

Revision and response to Reviewers comments for the manuscript entitled “Solar ultraviolet radiation in Pretoria and its relations with aerosols and tropospheric ozone during the biomass burning season

Reviewer 1 (R1)

The paper is better now in a lot of points but still has flaws which have to be corrected to allow publication.

Response: We thank the Reviewer for all of the comments that have helped us to improve the manuscript. We have addressed all the Reviewer’s comments in the responses below.

Details:

  1. Title: "relations to".

Response: The title of the manuscript has been revised to “Solar ultraviolet radiation in Pretoria and its relations to aerosols and tropospheric ozone during the biomass burning season”

  1. Abstract: Line 26: provide numbers from table 1 (aerosol up to 13% and ozone up to 1%).

Response: Numbers from Table 1 have been included in the Abstract.

  1. Keywords: add "ozone; UV" (TUV is too special).

Response: The keywords ozone and UV have been included.

  1. Eqn 2, line 159: please correct: numerator: times MED[h^-1], denominator:3600[s/h] (how it was written it is nonsense, there was a misunderstanding).

Response: Equation 2 has been corrected in the manuscript.

  1. Line 204ff: zonal average in the stratosphere or down to surface? What is done in the troposphere? Are the average sonde data used? This is not clear from line 197. That they are not used appears too late in the text (line 368).

Response: Clarification has been made in the manuscript to indicate which data sets were used in the troposphere and stratosphere, respectively (lines 207 and 209).

  1. Line 237: this holds for the lower troposphere only.

Response: This has been indicated in the manuscript.

  1. Line 243: better skip the word "trends" here (in the subtitle "climatology" was better).

Response: The subtitle has been revised.

  1. Line 254: "maxima"

Response: This has been corrected in the manuscript.

  1. Figure 3c and d and text lines 283ff: there must be something wrong, at least in January to March. It cannot be that lapse rate tropopause (same as in part b in contrast to caption?) and ozone tropopause differ so much. Check how many data points are there in each month. Are there bad profiles? If you cannot solve this, better skip this part since it appears to be not used later, maybe keep a remark in the text.

Response: The SHADOZ program does quality control on the data but after 2012, there are data gaps and it was not always possible to conduct an ozone sounding. Figure 3c and d have been removed and a remark has been added to the text.

  1. Line 326: Not 11 to 13h? Or is the time zone off from true solar local time? [just a comment]

Response: The highest daily UVI is between 12h-14h as the time zone for the entire South Africa is UTC+2. Cape Town is situated at a high latitude relative to most other towns and cities in South Africa, however, South Africa does not apply different time zones to account for these latitudinal differences.

  1. Line 363: skip "."

Response: This has been corrected in the manuscript.

  1. Figure 5: Don't compare apples and oranges. Leave out the lines with average observed all-sky data (?, not indicated in caption!) and use the clear-sky subset for observed averages as shown in Figure 4 but for the time period in the caption to support what is written in the text. Part a and b should have the same size.

Response: The data used for Figure 5 have been checked and the observed clear-sky data were used in Figure 5. This has been indicated in the legend. Figure 5a and b have been resized.

  1. Table 1: "(%) between modelled"? You may skip the last column and focus on September in the text. Footnotes are twice!

Response: The correction to Table 1 has been made and the text relating to the table has been revised.

  1. Line 439: typo

Response: This correction has been made and the entire manuscript has been checked for spelling and grammar.

  1. References: Line 557: typo

Response: This correction has been made in the manuscript.

  1. Line 576: URL?

Response: This correction has been made in the manuscript.

  1. Line 589: An organization has no first name.

Response: This correction has been made in the manuscript.

  1. Line 590: Is there an URL?

Response: There is no URL. The data were obtained directly from the South African Weather Service.

  1. Line 622: Reference incomplete, provide at least "AFCRL-68-0153" to enable the reader to find this technical report.

Response: The additional information has been included in the reference.

This manuscript is a resubmission of an earlier submission. The following is a list of the peer review reports and author responses from that submission.

Round 1

Reviewer 1 Report

The paper provides a lot of references to UV and aerosol but unfortunately the presented results were computed with too many and often not necessary simplifications or inconsistencies. The consequences of this are addressed in cited references. Measured ozone profiles and total ozone for the region are used in an odd and confusing way for the calculations.

Also the text requires improvements because it is often confusing or trivial. Don't use too many acronyms, RD for example is superfluous.

Details:
Line 35/113: Use a consistent boundary between UV-B and UV-A. It flips between two values in the text.

Line 42: Cause and effect appear to be messed up.

Lines 43/50: Biomass burning has a large anthropogenic contribution.

Line 58f/Fig.3: Mention here or later the seasonal variation of ozone due to the Brewer Dobson circulation in the stratosphere which causes enhanced downward transport in spring (September).

Line 62: Origin of pollution from biomass burning?

Line 88: Explain SDA or give a reference, the cited references do not discuss that.

Line 104: This assumption is in contradiction to Fig.3, include at least a seasonal dependency.

Eqn. 1: Check units, there appears to be something wrong.

Line 135: If there are hourly data you should have the diurnal cycle of UV, not only noon. You should also provide daytime average or the daily dose.

Line 143ff: Zenith angle is important but also the vertical profiles of aerosol. It matters at which altitude the biomass burning plume is. Mention the assumptions and the induced uncertainty.

Line 151ff: The ozone data of section 2.2 should be used for the troposphere and lower stratosphere, and AURA MLS for above, as a time series (2012-2018?), maybe with monthly averages if data are too sparse. The word 'climatology' is confusing here. It is better to analyse the years individually and do averaging later.

Line 156: Is that still valid? The constant SSA is in contradiction to the references and might be at least partially  responsible for the differences in Fig.4. Is there information on the origin of air masses and the aerosol type?

Line 159: This is a subset of the time span mentioned earlier and for first year there are no ozone data.

Line 200: Wrong reference?

Fig. 3: Include line with the subset used for the calculation of UV (with SD).

Line 222: Effect of filters? Change with time?

Line 224: Trivial.

Fig.5: Modify labels of time axis and caption concerning the used years. The observed average in part b appears to be dominated by outliers due to clouds out of scale, please check, there must be something wrong (and maybe in Fig. 4 as well). Maybe show the individual years in different colors. Address the dip in the biomass burning season.

Line 253ff: What is the background? You should better separate only the biomass burning aerosol and ozone effect, not for example the effect of seasonal transport from the stratosphere or the pollution from traffic and power plants, to be consistent with the abstract. If you are not able to do that show at least the whole seasonal cycle.

Line 287: Not only solar noon, please.

Line 292: Separate the seasonal effects of biomass burning and stratospheric transport in the text. If possible discuss also the NO2 impact separately to be more novel.

Line 295: Please read the provided references as help for analysis.

Reviewer 2 Report

The manuscript "The effects of tropospheric aerosols and ozone on solar ultraviolet radiation over Pretoria, South Africa”  by David Jean du Preez presents a study that is aimed to evaluate the impact of tropospheric aerosols loading and tropospheric ozone concentration on solar UVR over Pretoria during the biomass burning season, using  ground measurements and modelled data.

The language used in the manuscript is often difficult to understand, thus an extensive editing of English language and style is required. In many parts, the text is composed by short fragmented sentences with an insufficient insight into the issues discussed, turning into a low quality in the description of the study. Furthermore, the datasets and methods used in this study are not adequately explained and the description needs to be improved. Some of the statements are incorrect and need to be reformulated. More accuracy is needed both in the exposition and in the scientific analysis of the study.

Reviewer 3 Report

The paper by Preez et al. presents an interesting case of UVR levels at ground level and the effects of biomass burning from July to October in S. Africa. The results are interesting. I have a minor comment but important for the introduction. In line 3 there is a reference (No 2) to support the attenuation by ozone sulfur dioxide aerosols and clouds without quoting the basic paper on it by Bais et al., "Spectral measurements of solar UVB radiation and its relations to total ozone, SO2, and clouds", Geophys. Res., 98, D3, 5199-5204, 1993. I think that paper should be added in the introduction. With only this minor comment and perhaps a fine spell check the paper can be accepted in its present form.